# Integrated Stress Response (ISR) Pathway: Unraveling Its Role in Cellular Senescence

**DOI:** 10.3390/ijms242417423

**Published:** 2023-12-13

**Authors:** Alexander Kalinin, Ekaterina Zubkova, Mikhail Menshikov

**Affiliations:** 1National Medical Research Centre of Cardiology Named after Academician E.I. Chazov, 121552 Moscow, Russia; alexanderpkalinin@gmail.com (A.K.); cat.zubkova@gmail.com (E.Z.); 2Faculty of Fundamental Medicine, Lomonosov Moscow State University, 119991 Moscow, Russia

**Keywords:** stress response, ISR, ATF4, Nrf2, senescence, metabolism, SASP, cellular mechanisms

## Abstract

Cellular senescence is a complex process characterized by irreversible cell cycle arrest. Senescent cells accumulate with age, promoting disease development, yet the absence of specific markers hampers the development of selective anti-senescence drugs. The integrated stress response (ISR), an evolutionarily highly conserved signaling network activated in response to stress, globally downregulates protein translation while initiating the translation of specific protein sets including transcription factors. We propose that ISR signaling plays a central role in controlling senescence, given that senescence is considered a form of cellular stress. Exploring the intricate relationship between the ISR pathway and cellular senescence, we emphasize its potential as a regulatory mechanism in senescence and cellular metabolism. The ISR emerges as a master regulator of cellular metabolism during stress, activating autophagy and the mitochondrial unfolded protein response, crucial for maintaining mitochondrial quality and efficiency. Our review comprehensively examines ISR molecular mechanisms, focusing on ATF4-interacting partners, ISR modulators, and their impact on senescence-related conditions. By shedding light on the intricate relationship between ISR and cellular senescence, we aim to inspire future research directions and advance the development of targeted anti-senescence therapies based on ISR modulation.

## 1. Introduction

In 1961, Hayflick and Moorhead discovered that human diploid cells under in vitro passaging lose their proliferative capacity after a certain number of subcultivations [1]. Later Hayflick further defined this phenomenon and introduced the term senescence, which characterizes irreversible cell cycle arrest [2]. Since then, the physiology of senescent cells has been extensively studied, revealing a spectrum of cellular senescence types beyond replicative senescence. Developmental senescence (DS) is a programmed process crucial for embryonic development; its deregulation can lead to developmental abnormalities. Oncogene-induced senescence (OIS) acts as a natural anti-cancer mechanism, halting the proliferation of potentially cancerous cells and tagging them for immune detection. Therapy-induced senescence (TIS) occurs when treatment-induced cytostasis does not kill cells but leaves them functionally exhausted, leading to TIS. Lastly, oxidative stress-induced senescence (OSIS) arises from exposure to excessive amounts of reactive oxygen species (ROS) as well as ROS-generating agents or conditions [3,4,5,6].

These various forms of cellular senescence illustrate the multifaceted nature of this phenomenon.

Senescent cells are known to accumulate with age and contribute to the development of various diseases [7,8,9,10,11]. Different tissues are more or less affected by the accumulation of senescent cells. For instance, the pancreas, brain, and lung tissue are most susceptible to senescence [11]. In experimental mouse models, the elimination of senescent cells has shown beneficial effects on the course of several pathological conditions [12,13]. This has opened the possibility of developing potential anti-senescence compounds for clinical use. However, the current absence of specific markers for senescent cells makes it difficult to design safe and simultaneously selective anti-senescence drugs [14]. Therefore, a more comprehensive study of cellular senescence pathophysiology is needed to overcome this obstacle.

One mechanism whose role in the development of cellular senescence remains poorly understood is the integrated stress response (ISR). The ISR pathway is activated during several stressful conditions, such as amino acid deprivation and endoplasmic reticulum stress, and it exerts fine regulation over the cell’s translational and expression profiles [15]. Since entering a senescent state is considered a form of cellular stress and the ISR pathway is responsible for regulating the translational apparatus of the cell during stressful conditions, we propose that ISR signaling may not just indirectly influence the development of senescence but also play a central role in controlling this process.

In this review, we will not only provide a comprehensive summary of the most current scientific literature regarding the relationship between the ISR pathway and cellular senescence but also offer perspectives on potential avenues for future research. This review aims to encourage further exploration and a deeper understanding of the role of ISR as a regulatory mechanism in cell senescence.

## 2. An Overview of Cellular Senescence

Senescence is an elaborate cellular mechanism, aimed at immune clearance and preventing the proliferation of damaged, potentially cancerous, and mostly inflammatory cells. This process is triggered by various stressors and is characterized by several hallmarks, including irreversible cell cycle arrest accompanied by increased expression of cell-cycle regulating proteins like p16, p21, and p53; accumulation of phosphorylated histone 2AX (γH2AX) within senescence-associated heterochromatin foci (SAHF) and increased beta-galactosidase (SA-βGal) activity. Additionally, senescent cells undergo changes in their secretion patterns, leading to the development of a pro-inflammatory senescence-associated secretory phenotype (SASP) [16]. During the physiological response to damage, some cells may enter a senescence state and acquire a SASP phenotype that serves the purpose of “sensitizing” nearby cells to the presence of pathological foci within a tissue, thereby promoting tissue regeneration and further signal transduction to recruit immune cells to the injury focus and its repair [17]. SASP composition varies from cell to cell but most commonly includes the pro-inflammatory cytokines IL1 and IL6, chemokines IL8, CXCL1, CXCL2, CXCL3, MCP1, MCP2, MCP4, MIP1α, MIP3α, matrix metalloproteinases and growth factors [18]. Despite the widespread use of all the aforementioned hallmarks, there is currently no exclusive marker for senescent state. Consequently, the determination of whether specific cells are senescent relies on several non-exclusive features associated with senescence [14]. Moreover, certain senescent cells may not possess any of these hallmarks, therefore, identifying the senescent state of such cells may rely on the presence of less common senescence-associated factors, such as cytosolic double-stranded DNA or the urokinase-type plasminogen activator receptor [19,20,21,22].

While all the aforementioned markers: p16, p21, p53, phosphorylated histone γH2AX, SAHF, and beta-galactosidase staining—are widely used to identify senescent cells, none can unambiguously identify all such cells. For instance, SA-β-gal activity increases in senescent cells, but also under conditions unrelated to senescence, such as in confluent quiescent cells or cells experiencing serum starvation. Criticisms of SA-β-gal include its inducibility by differentiation and oxidative agents [23,24], its prevalence in high-density cultures, and its absence in certain tissues irrespective of donor age [25]. It is now understood that SA-β-Gal is lysosomal in origin, arising from increased lysosomal biogenesis, which partly reflects a rise in lysosomal mass, as detailed by Kurz et al. 2000 [26]. This is paralleled by the upregulation of the lysosomal β-D-galactosidase encoding gene GLB1 [27]. Thus, increased SA-β-gal staining could also reflect an alteration in lysosomal number or activity in non-proliferating cells. This variability suggests that increased SA-β-gal staining does not consistently correlate with cellular senescence, indicating that other factors may contribute to its expression.

Markers such as p16, p21, SAHF, and γ-H2A.X are primarily involved in fundamental cellular processes like cell cycle regulation and DNA damage signaling. Their specificity varies depending on the cell type and senescence stimuli [28,29], which precludes their use as exclusive markers of cellular senescence.

The existing literature offers mixed views on the role of p21 in senescence. While some studies emphasize its significance, others indicate that senescence can occur without p21 upregulation. For example, cellular senescence is possible in primary fibroblasts from p21 knockout mice, and resistance to senescence is observed in certain mouse hepatic tumor cells despite p21CIP1/WAF1 expression [30].

Additionally, p16INK4a and p21 may mark different cell populations within the same tissue, suggesting distinct senescence pathways. For example, in obese mice, a significant number of p21-expressing cells were found in visceral adipose tissue, including preadipocytes, endothelial cells, and macrophages, but not p16INK4a-expressing cells [31]. This observation supports the idea that p16INK4a- and p21-dependent senescence are separate and independent pathways. Furthermore, eliminating p16INK4a-expressing cells in aged animals has led to mixed outcomes, including both potential benefits and detrimental effects such as liver fibrosis and compromised wound healing [32,33,34]. Moreover, p16INK4a expression can be induced in non-senescent macrophages in response to physiological stimuli [21].

These nuances highlight the complexity of employing p16INK4a and p21CIP1/WAF1 as universal markers for senescence. The variability of their roles and expression across different cellular contexts, along with the fact that they are not exclusively expressed in senescent cells, are key reasons why these markers cannot conclusively identify all senescent cells.

The situation is further complicated by the fact that senescent cells are a highly heterogeneous population, as their phenotype depends on various spatiotemporal parameters [35]. Senescent cells fulfill a dual function in the physiology of organisms. On the one hand, they are necessary for normal embryogenesis and tissue regeneration [3,36,37]. On the other hand, the accumulation of senescent cells underlies the chronicity of various pathological conditions and so-called inflammaging, low-grade systemic inflammation, which progresses with age often without an infection in its pathogenesis [38]. Moreover, senescent cells may play a crucial role in the development of age-related diseases such as type 2 diabetes mellitus (T2DM), osteoarthritis, Alzheimer’s disease, and osteoporosis [7,8,9,10].

There is a hypothesis, that a transient existence of senescent cells in tissues and organs promotes their regeneration and resolution of inflammation. However prolonged survival of senescent cells mediates negative alteration of their phenotype and accumulation in tissues with the formation of a local inflammatory focus that later spreads its deleterious effect to other organ systems [39]. The transient existence of senescent cells in a young organism is ensured by a healthy immune system, which performs timely and efficient clearance of these cells. Conversely, in immunodeficient and elderly individuals can be observed senescence of the immune system itself—immunosenescence, which serves as the primary cause of the pathological accumulation of senescent cells [40]. Thus, these patients are in dire need of drugs aimed at suppressing the functioning of senescent cells. Senotherapeutics, a rapidly growing class of chemicals with anti-senescence effects, may become such drugs [41]. These compounds are categorized into two groups: senolytics, designed to eliminate senescent cells, and senomorphics, aimed at mitigating the pathogenic phenotype of senescent cells. Several FDA-approved drugs used in the treatment of various diseases have been found to exert an anti-senescent effect. For instance, metformin, a commonly used hypoglycemic drug for T2DM patients, has been shown to alleviate adipose tissue stem cells (ASCs) senescent state by potentiation of 5′AMP-activated protein kinase (AMPK) thereby partially restoring mitochondrial metabolism and differentiation potential of ASCs [42]. The effects of metformin have also been demonstrated in many other cell types. For example, Chen and colleagues have shown that metformin’s anti-senescent effect on nucleus pulposus cells relies on the activation of AMPK and autophagy [43]. The JAK1 and JAK2 kinase inhibitor Ruxolitinib, commonly used in the therapy of myelofibrosis and polycythemia vera, has been shown to suppress the SASP of senescent preadipocytes. This alleviates both systemic and local inflammation in adipose tissue in old mice [44]. However, it is important to note that these drugs are not primarily designed to target senescence, and their off-target use may come with potential side effects, raising concerns about their safety and specificity in eliminating senescent cells. For instance, low doses of metformin have been found to induce senescence in hepatocellular carcinoma cells (HCCs) [45]. Additionally, patients treated with Ruxolitinib may have an increased vulnerability to opportunistic infections and viral reactivation [46,47].

The pursuit of effective and safe senotherapeutic agents remains an ongoing challenge, primarily due to our limited understanding of cellular senescence mechanisms and the absence of specific markers for senescent cells. This enduring challenge has prompted a surge in research efforts over the past decades, driven by the need to unravel the detrimental impact of the growing burden of senescent cells which accompanies the aging process and exacerbates the chronicity of various diseases. Transgenic mouse models such as p16INK4a-INK-ATTAC and p16-3MR have been developed, enabling the specific elimination of p16-positive cells, and tracking their distribution in living organisms [12,34]. The creation of the metabolized in an organism prodrug SSK1 allows for targeted removal of SA-βGal-positive cells [48]. Omics technologies have been employed to compile signatures of various senescent cells [49,50,51]. These senescence-specific tools and databases allow researchers to precisely assess the role of senescent cell burden in their experimental models.

Given that cellular senescence contributes to the progression of various human diseases, understanding its mechanisms can significantly enhance the development of pharmacological therapies for these conditions. Therefore, in-depth investigations of specific signaling pathways in the pathophysiology of senescence are warranted. Among these pathways, the integrated stress response stands out as an intriguing candidate, as it responds to various stressors, and senescence itself is intricately linked with stress however ISR remains relatively understudied in its connection to senescence. In the following sections, we will explore the molecular mechanisms of the ISR.

## 3. Molecular Mechanism of Integrated Stress Response

The integrated stress response is a comprehensive and highly evolutionarily conserved mechanism that governs cell fate under extreme conditions [15]. ISR’s primary goal is to maintain the cell’s bioenergetics status until the stress is resolved. This is achieved by suppressing global protein synthesis (with a median 5.4-fold decrease of ribosomal occupancy, according to one estimate), Simultaneously, it initiates the selective synthesis of specific transcription factors such as ATF4 and essential enzymes crucial for cell survival [52,53].

The initial step of ISR is the activation of one or more kinases responsible for phosphorylating the eukaryotic translation initiation factor subunit alpha (eIF2α) [54]. These kinases are activated by a wide range of stressors including amino acid starvation (GCN2); endoplasmic reticulum (ER) stress (PERK); exposure to viral double-stranded RNA (PKR); mitochondrial stress and iron ions deficiency leading to heme depletion in erythroid cells (HRI) [54,55,56,57]. The hallmark of this pathway lies in its ability to integrate various signaling pathways, all converging on the phosphorylation of the same amino acid of translation initiation factor eIF2α, thus earning it the name ‘integrated stress response’.

The translation initiation factor eIF2, composed of α, β, and γ subunits, together with the initiator methionyl-tRNA (Met-tRNAi^Met^) and GTP, forms the ternary complex (TC) [58]. Subsequently, the TIC, along with other initiation factors—eIF1, eIF1A, and eIF3, binds to the 40S ribosomal subunit thereby forming a 43S preinitiation complex (PIC) [59].

Under normal unstressed conditions, the PIC with the involvement of eIF4 (A, B, E, F, G) further binds to the m7G cap structure located at the 5′ end of most cellular mRNAs [59]. Following the recognition of the initiation codon and the formation of the 48S initiation complex, eIF2 exhibits its GTPase activity hydrolyzing GTP with the assistance of the GTPase-activating factor eIF5 and the guanine nucleotide exchange factor eIF2B, thereby inducing the partial dissociation of eIF2-GDP from Met-tRNAi^Met^ [59]. Subsequently, initiation factors eIF1, eIF5, eIF2-GDP, and eIF3 dissociate, allowing for the attachment of the large 60S ribosomal subunit [59]. Once this step is complete, the translation initiation process is considered finished, and the 80S ribosome is ready for polypeptide synthesis.

Under stressful conditions, events may unfold differently. As mentioned above, following translation initiation, eIF2-GDP is released from the complex with the ribosome. To enable eIF2 to participate in translation again, it is necessary to replace GDP with GTP, a process regulated by the specialized guanine nucleotide exchange factor eIF2B [59]. Stress-induced kinases, namely HRI, PERK, PKR, and GCN2, phosphorylate the alpha subunit of eIF2 at serine 51 (p-eIF2α), and p-eIF2α further functions as an inhibitor of eIF2B [60], preventing the exchange of GDP for GTP. This results in the rapid depletion of active ternary complexes and a global reduction in protein synthesis. CAP-dependent translation virtually ceases, whereas the translation of mRNAs containing a short upstream open reading frame (uORF) in their 5′ untranslated regions (5′UTR) increases [61]. It is noteworthy that approximately 50% of human and mouse mRNAs contain at least one uORF [62], but ISR selectively upregulates translation for only a subset of these uORF-containing transcripts [63]. Among the upregulated proteins, notable targets include the aminoacyl-tRNA synthetase EPRS, the positively charged amino acid transporter CAT1, the negative regulator of the ISR signaling phosphatase GADD34, and transcription factors ATF4, ATF5, CHOP, CEBPA, and CEBPB [64]. Additionally, the transcription factor ATF3 may also be upregulated during ISR [65].

The integrated stress response serves as a critical and highly conserved cellular mechanism that orchestrates the cell’s response to extreme conditions. It efficiently modulates protein synthesis to adapt to various stressors, while selectively promoting the translation of specific transcripts crucial for cell survival. However, it is quite obvious that ISR signaling cannot manage every stress-inducing situation, as that would render cells immortal. More intriguingly, can ISR signaling subtly sense the current state of a cell and determine its fate? It can be assumed that an ISR outcome is not uniform and varies depending on the nature, duration, and intensity of exposure to a stressor. To comprehensively appreciate the significance of the ISR machinery in a cell, it is essential to investigate the diverse outcomes that ISR can induce. The main components of the ISR machinery are shown in Figure 1.

## 4. Distinct Outcomes of ISR

The integrated stress response holds the power to determine cell fate depending on the nature, duration, and intensity of the stressor. One of the most intriguing properties of ISR signaling is its potential to either return a cell to its physiological state or lead it toward cell death. In the subsequent sections, we review the distinctive characteristics of various ISR outcomes.

When stress is relieved, the ISR kinases are deactivated due to the absence of triggers allowing the cell to return to its physiological state through the de-phosphorylation of p-eIF2α. This process is mediated by the protein phosphatase 1 (PP1), which acts as a catalytic subunit in complexes with either CReP or GADD34 [66]. CReP is constitutively expressed providing a baseline level of eIF2α de-phosphorylation, while GADD34, whose translation is permitted by ISR signaling and whose gene expression is upregulated by ATF4, is an inducible factor and a part of the ISR negative feedback loop [67,68,69]. De-phosphorylation of p-eIF2α enables eIF2 to participate in translation again, this is the most favorable and physiological ISR outcome for a cell, as ISR signaling preserves as many cellular components as possible in their native state and eliminates those that were damaged or dysregulated.

However, under conditions of prolonged or excessive stress, the ISR pathway directs a cell to undergo programmed cell death and apoptosis [70]. A major effector of the apoptotic outcome is the ISR transcription factor CHOP, which can upregulate pro-apoptotic proteins such as BIM, PUMA, and death receptor 4 (DR4) and 5 (DR5) while simultaneously downregulating anti-apoptotic factors like MCL1, BCL-XL, and BCL2 [71,72,73,74]. The ISR key transcription factors ATF3 and ATF5 have also been reported to promote apoptosis [75,76]. However, it is important to keep in mind that since CHOP, ATF3, and ATF5 factors function only as a dimeric complex, the products of their transcriptional activity are largely influenced by their dimerization partner. This fact explains the existence of studies demonstrating the anti-apoptotic effects of ATF3 and ATF5 [77,78].

GADD34 phosphatase is another significant regulator of apoptotic outcomes. It mitigates apoptosis by both attenuating the overall ISR pathway and promoting the stabilization of the BCL-2 family member, myeloid cell leukemia 1 (MCL1) protein. This dual action of GADD34 contributes to enhanced cellular autophagy [79,80,81].

However, in certain conditions, such as exposure to lipopolysaccharide, autophagic death of A549 cells can be mediated through the induction of ER stress and the subsequent activation of the PERK-eIF2α-ATF4-GADD34 signaling pathway [82]. Excessive stress may lead to persistent activation of the ISR signaling, resulting in a substantial upregulation of GADD34. This, in turn, can mediate prolonged autophagy activation and potential subsequent autophagic cell death. Verfaillie et. al. unveiled a novel role of PERK kinase as an integral component of mitochondria-associated ER membranes (MAMs), facilitating the propagation of Ca^2+^ and ROS signaling between the ER and ER-associated mitochondria [83]. Knockout of the PERK gene partially prevented the spreading of ROS signaling to mitochondria, protecting them from oxidative damage, but also causing disruption of the Ca^2+^ signaling pathway and ER morphology in a cell [83]. Thus, PERK can exert a proapoptotic effect not only by upregulating CHOP through the activation of the ISR pathway but also by propagating ROS signaling to mitochondria [83]. There are also reports implicating HRI, PKR, and GCN2 in the apoptotic outcome, but the proapoptotic effects of these kinases are primarily limited to the activation of the ISR signaling pathway [84,85,86]. Therefore, the principal mediators of ISR apoptotic outcome are the transcription factors ATF3, ATF5, and especially CHOP, while PERK and GADD34 less contribute to proapoptotic effects.

Another outcome of ISR is an alteration in cellular differentiation status. D’Aniello and colleagues, using mouse embryonic stem cells (ESCs), described a GCN2-eIF2α-ATF4 autoregulatory loop. This loop involves the activation of ISR signaling mediated by a deficiency in the L-proline amino acid, preserving ESCs in their native state, whereas L-proline sufficiency led to attenuation of ISR along with the downregulation of ATF4, resulting in a mesenchymal-like transition of ESCs [87]. Mielke and colleagues demonstrated that the phosphorylation of eIF2α is required in the late stage of B-lymphocyte maturation [88]. This eIF2α phosphorylation helps maintain myosatellite cells (MuSCs) in a quiescent state and enables their self-renewal. In contrast, p-eIF2α de-phosphorylation mediates the induction of the myogenic differentiation program and reduces the proliferative capacity of MuSCs [89].

ATF4 plays a crucial role in the proliferation and differentiation of various cells, however, this effect of ATF4 is often considered outside the context of ISR signaling [90,91,92,93]. Since ISR transcription factors like ATF4, ATF5, CHOP, CEBPA, and CEBPB, contain a basic leucine zipper (bZIP) domain, they can interact with both other bZIP-containing proteins as well as members of the AP-1 family. This interaction allows them to form homo- or heterodimers, which are capable of binding to DNA, thereby creating a unique cellular expression profile.

Notably, there are more than 40 ATF4 binding partners, which can potentially affect the transcription activity of ATF4 [94]. This suggests that the ISR program may be defined by the cellular concentration of these ISR transcription factors and their interacting partners.

Unfortunately, none of the aforementioned reports describing the possible outcomes of ISR outline the composition of dimer complexes in which ISR transcription factors act. The example of ATF4, which appears to play a significant role in regulating cellular differentiation, underscores the importance of identifying the dimerization partners of ISR transcription factors. This importance is further emphasized by the fact that seemingly identical signaling pathways can lead to extremely distinct outcomes. Therefore, to achieve a comprehensive understanding of how ISR can lead to such diverse outcomes, it is essential to thoroughly examine the roles played by ISR effectors within a living cell.

## 5. ISR in Cellular Metabolism

### 5.1. ISR and Autophagy

Cellular autophagy is a highly conserved catabolic process responsible for the recycling of dysfunctional biomolecules and organelles as well as maintaining intracellular levels of essential monomers and substrates needed for anabolism under conditions of their deficiency [95]. Autophagy is initiated by inhibiting the Mechanistic Target of Rapamycin Complex 1 (mTORC1), followed by the activation of the ULK1 complex, autophagosome formation, its fusion with the lysosome and subsequent enzymatic degradation of the autophagolysosomal contents [95].

Interestingly, even though the activation of ISR leads to the suppression of global protein synthesis, this pathway can be considered anabolic since it involves some form of protein synthesis rather than degradation. This observation leads to the hypothesis that autophagy and ISR are intricately linked, mutually influencing each other. Indeed, B’chir and colleagues showed that autophagy progression under conditions of amino acid starvation and ER stress requires activation of the ISR pathway, resulting in the upregulation of its effectors, including the transcription factors ATF4 and CHOP [96]. Specifically, the ATF4-CHOP complex mediates transcriptional activation of the autophagy-related genes such as NBR1, ATG7 and SQSTM1, while the transcription of ATG10, GABARAP and ATG5 depends on both ATF4 and CHOP, but not their dimeric complex, Furthermore, the transcription of other autophagy genes like ATG16L1, MAP1LC3B, ATG12, ATG3, BECN1 and GABARAPL2 is ATF4-dependent but does not require CHOP involvement [97].

Yang and colleagues have shown that the mRNAs of autophagy-related proteins ATG5 and ATG13 include upstream open reading frames (uORFs) that suppress their CAP-dependent translation. They also compiled a list of 32 autophagy-related proteins with mRNAs that potentially contain active uORFs [97]. This suggests that the integrated stress response may not only upregulate the expression of autophagy proteins but also enable the autophagy progression as a whole.

ATF4, in turn, plays a pivotal role in cellular anabolism by promoting the transcription of various enzymes (ASNS, PSAT1, PHGDH, PSPH, SHMT2, MTHFD2, PYCR1) and amino acid transporters genes (SLC1A4, SLC7A1, SLC7A5) [98]. Interestingly, the glucocorticosteroid dexamethasone suppresses ATF4, whereas insulin stimulates ATF4 by activating the mTORC1 complex [98]. Subsequently, several research groups confirmed that mTORC1 activates the translation of ATF4 [63,99,100]. Park and Selvarajah suggest that upregulation of the ATF4 translation is mediated by eukaryotic translation initiation factor 4E-binding protein 1 (4E-BP1), which in turn is activated by the mTORC1 complex [99,100]. Thus, ATF4 has the capacity to stimulate both anabolic and catabolic processes in the cell, further highlighting the pleiotropic nature of its effects.

GCN2 kinase is one of the key sensors of amino acid starvation [54]. Considering that autophagy is an adaptive cellular mechanism operating under conditions of nutrient starvation [101], it is reasonable to propose that GCN2 may play a significant role in activating autophagy during amino acid deprivation. Indeed, it has been demonstrated that GCN2 kinase is capable of initiating autophagy by suppressing the mTORC1 complex [102,103,104]. Notably, GCN2-mediated short-term activation of autophagy requires both the phosphorylation of eIF2α and the suppression of the mTORC1 complex, but it does not necessitate the activation of ATF4 [103]. It can be hypothesized that GCN2 activates the «early» autophagy by mobilizing the basal pool of existing autophagy proteins and their mRNAs that are already present in a cell. However, when this pool is depleted, the transcriptional activity of factors such as ATF4 and CHOP may become necessary for autophagy to proceed. Additionally, GCN2 can sustain autophagy through the activation of the p-eIF2α-ATF4 axis, thereby upregulating the stress response protein Sestrin2. Sestrin2 is required to maintain mTORC1 repression by blocking its localization to lysosomes [104].

The ISR pathway eventually leads to the activation of transcription factors ATF3 and ATF4, which, in turn, induce the expression of phosphatase GADD34. GADD34, in complex with PP1, dephosphorylates p-eIF2α, thereby terminating ISR and forming a negative feedback loop of ISR [65,69]. Intriguingly, GADD34 can stimulate cellular autophagy by indirectly inhibiting mTORC1 through the dephosphorylation of the phosphorylated form of tuberous sclerosis complex 2 protein (pTSC2), facilitating the assembly of the TSC1/TSC2 complex which subsequently inhibits mTORC1 [105].

Kapuy and colleagues built a control network model of cellular autophagy based on a comprehensive analysis of over 100 scientific papers [106]. The authors defined GADD34 as an “autophagy inducer” and CHOP as an “autophagy controller” in the context of ER stress [106]. Moreover, GADD34 both stimulates CHOP and inhibits mTORC1, while CHOP and mTORC1, in turn, suppress GADD34 activity [106]. During ER-stress GADD34 can activate cellular autophagy through both CHOP stimulation and mTORC1 inhibition. Therefore, CHOP downregulation does not lead to autophagic flux blockage, whereas depletion of GADD34 completely diminishes autophagy [106].

In summary, various components of the ISR pathway appear to have a significant role in cellular autophagy regulation: kinase GCN2 and phosphatase GADD34 activate autophagy and ATF4 and CHOP transcription factors stimulate the expression of autophagy-related genes. Given that GCN2 and GADD34 have opposite functions within the ISR pathway but the same effect on autophagy, it is conceivable that the ISR integrates various stress signals to finely tune cellular metabolism according to the specific circumstances. 

### 5.2. ISR and Mitochondrial Homeostasis

Mitochondria are the bioenergetic center of the cell and play a crucial role in metabolism [107]. The mitochondrial theory of aging, originally proposed by Denham Harman, postulates that mitochondria experience dysfunction with age, resulting in the excessive generation of reactive oxygen species (ROS). These ROS, in turn, mediate oxidative damage to biomolecules, ultimately contributing to the aging process [108]. Additionally, Wiley and colleagues characterized mitochondrial dysfunction as another trigger of cellular senescence, further underscoring the importance of maintaining mitochondrial health to combat both cellular senescence and aging processes [109].

Quirós and colleagues conducted a multiomics analysis, revealing, that the ISR functions as an adaptive mechanism during mitochondrial stress. Notably, ATF4 doesn’t solely exert its cytoprotective effect under stressful conditions by enhancing the transcription of genes like asparagine synthetase (ASNS) and phosphoserine phosphatase (PSPH); it also plays a broader role in maintaining mitochondrial homeostasis [110]. ASNS and PSPH catalyze the synthesis of the vital amino acids asparagine and serine, respectively. These amino acids are produced using precursors generated as products of the mitochondrial tricarboxylic acid cycle. Mitochondrial impairment, such as electron transport chain (ETC) dysfunction, can lead to a reduction in cellular asparagine, aspartate, and serine levels as well as NAD^+^/NADH ratios [111,112]. Interestingly, when ETC dysfunction occurs, cells reduce the uptake of exogenous serine while increasing its cytoplasmic synthesis [112]. This response may allow cells to overcome the blockade of mitochondrial serine metabolism through mass action, but only at high cytoplasmic serine concentrations [112]. Since mTORC1 senses metabolites to coordinate anabolic activity with the availability of biosynthetic precursors, it is suggested that the lack of arginine during ETC dysfunction impairs mTORC1 activity [111]. It can be assumed that autophagy, stimulated by mitochondrial dysfunction, utilizes impaired macromolecules and organelles while simultaneously serving as a source of monomers for synthesis through enzymes upregulated by ISR.

### 5.3. ISR and Mitochondrial Unfolded Protein Response

Mitochondrial unfolded protein response (UPR^mt^) and autophagy are crucial mechanisms for maintaining the quality and efficiency of the cellular mitochondrial network [113,114]. It is widely accepted that the activation of the ISR pathway is essential for UPR^mt^ [114]. Transcription factors ATF4, ATF5, and CHOP, upregulated by ISR signaling, play key roles in UPR^mt^ by stimulating the expression of proteases, chaperones, and metabolic genes, thereby promoting mitochondrial recovery and cell growth [114]. ISR-induced autophagy likely contributes significantly to the maintenance of mitochondrial homeostasis. This hypothesis finds support in a study by Condon’s group, in which they demonstrated that AMPK is involved in an early phase of adaptation to mitochondrial stress, mediating short-term inhibition of mTORC1. Meanwhile, HRI is involved in the cell’s adaptation to prolonged stress by triggering the ISR and ATF4-dependent upregulation of inhibitors of mTORC1 signaling, namely, the proteins Sestrin2 and Redd1 [115].

Whitney and colleagues demonstrated that under tunicamycin- or thapsigargin-induced ER stress, ATF4 is the key stimulator of Redd1 protein expression, and upregulation of ATF4 occurs through the PERK-peIF2α axis [116]. Fessler’s and Guo’s research groups go further, revealing a novel mechanism of HRI kinase activation triggered by mitochondrial perturbations: mitochondrial stress initiates OMA1-dependent cleavage of the large form of DELE1 protein into its smaller form. This smaller form can then exit the mitochondrion and activate HRI in the cytoplasm [55,57]. The significance of HRI-mediated mitochondrial quality control in vivo was underscored by Zhu and colleagues, who uncovered a protective role of mitochondrial stress-triggered HRI-eIF2α-ATF4 signaling in both fetal and adult mitochondrial cardiomyopathy [117].

In summary, the ISR machinery is intricately involved in the regulation of both UPR^mt^ and autophagy. Recent research has shed light on the central role of ATF4, the major effector molecule of ISR signaling, in stimulating UPR^mt^ and autophagy. This dual activation promotes enhanced mitochondrial networking and the capacity to clear dysfunctional organelles under stress conditions [118].

The ISR pathway, serving as a master regulator of cellular metabolism during stress, plays a pivotal role in activating both autophagy and UPR^mt^. Cellular senescence, characterized by the acquisition of a senescent phenotype, often occurs in response to stressful conditions such as oncogene activation or exposure to genotoxic agents [4,5,6]. Given that the ISR governs the translational machinery of a cell during stress, it strongly implies that ISR signaling is inevitably intertwined with the regulation of processes associated with cellular senescence.

## 6. Interrelationship between ISR and Cellular Senescence

The ISR signaling pathway can be divided into three main steps: initiation through phosphorylation of eIF2α by four different kinases; the ISR program execution through inhibition of global protein synthesis and selective translation of certain molecules; and ISR termination. Termination occurs either through stress resolution and subsequent dephosphorylation of p-eIF2α or by triggering cell death via apoptosis.

The initial step of ISR involves the activation of specific kinases, among them GCN2, which primarily responds to amino acid starvation [54]. Intriguingly, despite the well-documented detrimental effects of a high-protein diet, prolonged amino acid deprivation can result in cellular senescence [119,120]. Studies employing mouse embryonic fibroblasts (MEFs) have revealed that GCN2 activation due to amino acid deprivation leads to the p53-independent accumulation of p21, a cyclin-dependent kinase inhibitor, and senescence marker protein [121]. Similar observations were made in hepatocellular carcinoma cells by Missiaen and colleagues, where arginine deprivation induced GCN2-mediated upregulation of p21 expression, while p16INK4a and p27kip remained unaffected [122]. Interestingly, the inhibition of GCN2 using the GCN2iB small molecule during arginine deprivation induced high-grade senescence. This was characterized by a significant increase in the number of SA-βGAL-positive cells and elevated expression of SASP. Notably, these senescent cells became more susceptible to ABT-263 senolytic therapy [122].

It is crucial to emphasize that GCN2 kinase functions as a sensor of amino acid deficiency in a cell, serving as a vital cytoprotective mechanism. The upregulation of cell cycle inhibitors, mediated by GCN2, appears plausible. Cells prioritize survival over proliferation when confronted with starvation-induced stressors. They direct available cellular resources toward ensuring their survival rather than promoting cell division.

However, it is equally important to highlight that switching off GCN2 kinase can have profoundly negative consequences, leading to profound cell exhaustion during amino acid starvation. This exhaustion often results in the acquisition of a senescent phenotype, underscoring the pivotal role of GCN2 in maintaining cellular homeostasis. Further support for the importance of GCN2′s function comes from studies in which its inhibition resulted in the suppression of tumor growth. This indicates that even cancer cells deviate from their normal state when GCN2 kinase is downregulated, emphasizing the broader significance of GCN2 beyond cellular senescence [123,124].

PERK kinase, an ER stress sensor, is also implicated in senescence development. Research by Rajesh and colleagues using primary mouse and human fibroblasts demonstrated that the ablation of PERK or p-eIF2α results in premature senescence due to cellular ROS accumulation [125]. Furthermore, Zhang and colleagues elucidated a novel cGAS-STING-PERK-eIF2α signaling pathway and its connection to cellular senescence and fibrosis. An effector of innate immunity, cyclic GMP-AMP synthase is capable of binding to viral, bacterial, and mitochondrial DNA in a cytoplasm. This binding thereby triggers a pro-inflammatory signaling cascade, which in turn leads to activation of the NF-κB and IRF3 transcription factors and stimulates autophagy or, in some cases, promotes cell death [126]. Zhang and colleagues were the first to show that the stimulator of interferon genes (STING) interacts with the kinase domain of PERK, thereby activating PERK [126]. STING-PERK signaling marginally activates the ATF4-CHOP axis, suggesting the execution of a specific UPR-independent program [126]. Thus, cytosolic DNA is another activator of ISR in addition to heme deficiency, RNA viral infection, ER stress, and amino-acid starvation. Furthermore, the capacity of STING to induce PERK-eIF2α signaling is evolutionarily retained from anemone to fly, zebrafish, and humans and plays a pivotal role in damage-induced senescence [126]. Knockout of PERK in primary MEFs resulted in decreased expression of p21 and reduced number of senescent cells in the DNA-damage-triggered senescence model [126].

PKR kinase, which is activated by viral double-stranded RNA, phosphorylates eIF2α and is upregulated by the p53 protein, resulting in significant tumor growth suppression [127]. Additionally, PKR activates the IKK complex through phosphorylation, leading to the upregulation of the NF-κB signaling pathway, a major contributor to the senescence phenotype [128]. Li and colleagues demonstrated that PKR kinase also upregulates c-Jun N-terminal kinase (JNK), which suppresses the expression of the anti-senescent factor Sirtuin 1, thereby promoting the development of palmitate-induced senescence in human umbilical vein endothelial cells (HUVECs) [129]. However, the role of JNK in senescence development is a subject of debate, with some studies suggesting that basal levels of JNK protein are required to counteract senescence [130,131,132]. Therefore, ISR kinases have multiple functions beyond ISR signaling, and their roles in senescence are likely to depend on the specific cellular context. For example, PERK and PKR can activate glycogen synthase kinase, promoting the translocation of p53 from the nucleus to the cytoplasm and its subsequent proteasomal degradation [133].

It is intriguing that the effector of the ISR negative feedback loop, phosphatase GADD34, can also play a role in the regulation of cellular senescence. Upregulation of GADD34 promotes p53 activation and has been associated with negative dynamics of conditions such as cerebral ischemia and vascular atherosclerosis [134,135]. Additionally, GADD34 has the capability to upregulate the p21-pro-senescence protein and cell cycle inhibitor [136,137,138]. The presence of such a senescence-associated signaling mechanism seems plausible, as elevated levels of GADD34 often serve as a marker of persistent stress.

The ISR program entails suppression of global protein synthesis with subsequent selective translational activation of specific molecules, such as the transcription factors ATF3 and ATF4 [65]. Using the replicative senescence model of HUVECs, Zhang and colleagues determined the increased accessibility regions (IARs) of cell chromatin and found that the AP-1 family motifs were the most enriched in IARs, among which the ATF3 motif was the most significant [139]. According to the results of the study, knockout of the ATF3 gene mediates the downregulation of the p16 protein and reduction of the SA-βGAL signal in HUVECs. Meanwhile, overexpression of ATF3 leads to the excessive binding of ATF3 to IARs, thereby upregulating the IARs-associated genes, such as p21 and p16 cell cycle inhibitors genes [139]. On the other hand, using the Pseudomonas aeruginosa-induced senescence model of RAW264.7 macrophages, Zhao and colleagues showed that senescence is accompanied by the activation of ATF3, which functions as an oxidative stress-responsive cytoprotective element and anti-senescence factor [140]. Whereby, in the senescent macrophages, ATF3 knockdown results in aggravated senescence, and increased TNFα and IL6 expression but suppressed IL10, whereas ATF3 overexpression has diametrically opposite effects [140]. Kim and colleagues showed that acrylamide treatment of murine macrophages leads to the development of senescence, characterized by the accumulation of ROS in these cells and activation of the p38 and JNK pathways, which in turn upregulate the ATF3 factor that promotes ROS generation and upregulation of p53 and p21 pro-senescent proteins [141]. ATF3 can prevent MDM2-mediated destruction of the p53 pro-senescent protein by disrupting its ubiquitination, so the knockdown of ATF3 results in inefficient p53 induction and impaired apoptosis [142]. Using HCT116 colorectal cancer cells treated with the DNA damage inducer camptothecin, Taketani and colleagues showed that ATF3 acts as a co-transcription factor for p53. Both ATF3 and p53 are recruited to the DR5 gene promoter and interact to ensure effective transcription of the DR5 gene, thereby promoting apoptosis [143]. Since ATF3 binds to 40% of p53 target genes [144], this interaction may have a crucial role in enhancing cell death caused by genotoxic agents. Therefore, ATF3 is one of the p53 target genes under conditions of genotoxic stress and is also able to form a complex with p53, regulating the transcription of several genes. ATF3 in turn regulates the expression of several senescence-associated proteins. This makes ATF3 a master regulator of cell proliferation and thus of cellular senescence.

ATF4 is considered to be the most essential ISR transcription factor. Hematopoietic stem cells from young knockout mice (ATF4-/-) demonstrate the formation of an aging-related defective phenotype characterized by the accumulation of reactive oxygen species and downregulation of hypoxia-inducible factor 1-alpha, the lack of which in the cell leads to further aggravation of oxidative stress [145]. On the other hand, Liu and colleagues showed that expression of the cell cycle inhibitor p16 is upregulated by ATF4 during EPR stress-induced senescence in mouse renal tubular epithelial cells [146]. Another study also showed increased expression of p16 in MEFs upon activation of the HRI/p-eIF2α/ATF4 signaling pathway induced by exposure to hydroxyl radical [147]. We analyzed over 170 ATF4 target genes for alterations in their expression levels in senescent cells using the SeneQuest v6 (senequest.net accessed on 15 August 2023) and found that the most upregulated genes are CHOP, CEBPB, GDF15, SQSTM1, ANGPTL4, VEGFA, GADD45A and KDM6B [94]. The protein SQSTM1 is an autophagic adapter that acts as a cargo receptor for the degradation of ubiquitinated substrates and is upregulated, as mentioned before, by the ATF4-CHOP complex [96]. KDM6B is an inducible histone demethylase, it can modulate inflammatory responses, such as M2 polarization via STAT6, and direct the commitment of CD4+ T cells via the T-bet factors [148]. KDM6B is also able to upregulate p53 and INK4 box genes like p15 and p16 cell cycle inhibitors genes [148]. KDM6B can prevent cancer formation via OIS induction since RAS and p53 signaling stimulate KDM6B function [148]. Being a component of the DNA damage response, GADD45A maintains cellular stability by regulating certain nucleotide excision repair proteins, cell cycle regulators, protein kinases, and base excision repair proteins or, in case of excessive damage, GADD45A promotes apoptosis [149]. GADD45A can be activated by p53 and in turn, upregulates p21 cell cycle inhibitor through activation of the p38 MAPK pathway [149]. Despite the presence of GDF15, ANGPTL4, and VEGFA in a senescent phenotype, these factors play a beneficial role in tissue regeneration and homeostasis, so their effect can mostly be described as positive, regardless of their involvement in cancer progression and metastasis [150,151,152]. What’s even more intriguing is that the typical ISR transcription factors CHOP and CEBPB are also presented in the list, highlighting the need for a detailed description of these factors in the context of cellular senescence.

It is crucial to acknowledge a significant limitation in the studies mentioned: they examined the effects of ATF3 or ATF4 without taking into consideration that these transcription factors can only function in homodimers or heterodimers with other transcription factors, and ATF4 is unable to form highly stable homodimers [153]. Furthermore, research by Rodríguez-Martínez has revealed that many previous studies, that assumed the use of ATF4 homodimers, may have been working with impure ATF4-CEBPG heterodimers [154]. This suggests that the overall outcome of the ISR pathway may be contingent on the quantity or presence of specific ATF4-interacting proteins within a cell. For example, PDX1, which forms dimers with ATF4, is predominantly expressed in beta-cells of Langerhans islets and duodenal enterocytes, leading to a distinct and vital outcome of the ISR in these cells [155,156]. Therefore, to comprehensively assess the impact of the ISR pathway on the development of cellular senescence, it is imperative to accurately characterize the potential heterocomplex variants of the key ISR transcription factor ATF4.

## 7. Role of ATF4-Interacting Partners in Cellular Senescence

CCAAT/enhancer-binding proteins (C/EBPs) constitute a transcription factor family with six known members: CEBPA, CEBPB, CEBPG, CEBPD, CEBPE, and CHOP. These proteins play a pivotal role in regulating gene expression across various biological processes, including cell proliferation, growth, and differentiation. All members of this family share a common DNA-binding domain known as the basic leucine zipper (bZIP). C/EBP proteins can form homodimers and homotypic heterodimers by using the leucine zipper dimerization domain to interact with other members of the CREB/ATF subfamily [157]. In the following sections, we will explore the potential roles of these protein complexes in the context of senescence development. The complexity of the involvement of ISR transcription factors in cellular senescence is illustrated in Figure 2.

### 7.1. ATF4-C/EBP Complex

Horiguchi and colleagues have shown that the heterodimeric complex formed by ATF4 and CEBPA binds to the CARE (C/EBP-ATF response element) sequence in the promoter region of the CDKN2A gene [158]. CDKN2A encodes the negative regulators of proliferation p16INK4a and its alternative transcript p19ARF, and the ATF4-CEBPA complex effectively represses their transcription [158]. Notably, this repressive ability is unique to the heterodimer, as the homodimers of ATF4 or CEBPA have minimal effect on CDKN2A gene expression [158]. Huggins and colleagues observed that unlike interactions with CEBPB or CHOP, ATF4 dimerization with CEBPG mediates an antioxidant, anti-senescence effect during oxidative stress, amino acid starvation, and endoplasmic reticulum stress [159]. Conversely, the homodimeric complex of CEBPB inhibits cell proliferation and contributes to SASP formation [160]. However, the heterodimer formed by CEBPB and CEBPG counteracts these pro-senescence effects [160]. It is interesting to note that the LIP isoform of CEBPB can downregulate ATF4 [161]. In addition, Guan and colleagues have shown that during replicative senescence in MEFs, CEBPA can potentiate senescence-activated enhancers within topologically associated chromatin domains, contributing significantly to the development of SASP phenotype [162]. This is achieved through the upregulation of pro-senescence genes including CXCL1, CXCL5, CXCL15, IGFBP2, and IGFBP6 [162].

### 7.2. ATF4-CHOP(DDIT3) Complex

ATF4 is known to bind to the AARE1 and AARE2 motifs of the CHOP gene promoter and activate its expression [163]. Kaspar and colleagues suggest that CHOP works with CEBPB to prevent the over-activation of ATF4, thereby controlling the ISR machinery [164]. In MCF7 cells, it has been shown that ATF4, in cooperation with CHOP, contributes to the enhanced transcription of p21, which plays a crucial role in promoting cell survival during ER stress [165]. In alveolar epithelial cells from patients with idiopathic pulmonary fibrosis, Jing and colleagues observed markers of senescence and ER stress [166]. This was accompanied by the upregulation of CHOP, which is recognized as a marker of ISR [166]. Based on in vitro and in vivo models, the authors discovered that CHOP stimulates ROS generation and subsequently activates the NF-κB pathway, which is considered to be a major contributor to the development of SASP [166]. Therefore, according to a comprehensive analysis of the literature, it can be hypothesized that the dimerization of ATF4 with CEBPG has an anti-senescence effect, whereas ATF4 interaction with CHOP promotes a pro-senescence outcome.

### 7.3. ATF4-NRF2 (NFE2L2) Complex

The bZIP class of transcription factors is one of the largest and includes numerous subfamilies such as the aforementioned CREB/ATF and C/EBPs, as well as activator protein-1 (AP-1), the musculoaponeurotic fibrosarcoma family and the nuclear factor (erythroid 2)-like (NFE2) or Cap‘n’collar (CNC) subfamily. A member of the latter subfamily is the NRF2 protein (NF-E2 p45-related factor 2). NRF2 consists of seven domains, from NEH1 to NEH7. NEH1 contains the DNA binding and dimerization domain, which includes the CNC-bZIP region. NEH2 contains DLG and ETGE motifs that are critical for NRF2′s interaction with its negative regulator, Kelch-like-ECH-associated protein 1 (KEAP1). KEAP1 acts as an adapter protein for the cullin-based E3 ubiquitin ligase. When KEAP1 binds to NRF2, it leads to the proteasomal degradation of NRF2. However, during oxidative stress, cysteine residues in KEAP1 are oxidized, causing KEAP1 to dissociate from the complex. As a result, NRF2 is transported to the nucleus [167].

NRF2 acts as a key cellular protector during oxidative stress by activating the transcription of phase II detoxification enzymes. The primary function of these enzymes is to detoxify highly reactive intermediate metabolites generated by phase I reactions, which include oxidation, reduction, and hydrolysis reactions. Examples of these detoxifying enzymes include NAD(P)H quinone oxidoreductase 1, glutathione peroxidase, ferritin, and heme oxygenase [167]. Additionally, NRF2 can suppress both NF-κB signaling and the expression of several pro-inflammatory genes, including IL-1β and IL-6. The exceptional properties of NRF2 not only establish it as a key biomolecule in maintaining cellular redox balance, detoxifying xenobiotics, and biotransforming free radicals but also as a vital anti-senescence factor [168]. Moreover, compounds that activate NRF2 may soon form a distinct category of senotherapeutic agents [169]. Consequently, NRF2 has been recognized as the master regulator of the cellular oxidative stress response and as the key anti-senescence effector [168]. Furthermore, it is noteworthy that NRF2 can also be activated by endogenous cues, such as ER stress and disturbances in autophagy [168].

He and colleagues were among the first to demonstrate the possibility of ATF4 dimerizing with NRF2 in mammalian cells. They found that ATF4, in complex with NRF2, can bind to a cis-acting element named the Stress-responsive Element [170]. Subsequent works by Su, Reinke, and Poh confirmed that NRF2 dimerizes with ATF4. However, the detailed outcome of ATF4-NRF2 complex formation remains to be fully elucidated [171,172,173]. It is known that the PERK-kinase can phosphorylate NRF2 bound to KEAP1, mediating dissociation of the NRF2-KEAP1 complex and thereby NRF2 activation [174]. On the other hand, Sarcinelli and colleagues identified ATF4 as the primary player in NRF2 activation during ER stress and PERK-induced ISR. This is attributed to the capability of ATF4 to interact with the promoter region of the NRF2 gene, which contains a CARE sequence [175]. This finding was further confirmed in a recent study by Kress and colleagues, demonstrating that substantial NRF2 upregulation occurs through ATF4 activation via the ISR signaling as a whole, meaning it is not solely dependent on its trigger [176]. Intriguingly, CHOP undergoes proteasomal degradation upon binding to a supercomplex consisting of the COP9 signalosome and Keap1, along with Cullin 3 [177]. This suggests the potential for crosstalk between the NRF2/Keap1 and CHOP/Keap1 complexes within the cell, potentially influencing each other’s activities and affecting cellular processes related to oxidative stress, inflammation, and senescence. However, further research is needed to validate and elucidate the mechanisms of this potential interplay.

### 7.4. ATF4 and Other bZIP Transcription Factors

ATF4 has been demonstrated to form dimeric complexes with other bZIP transcription factors such as FOS, MAF, FRA1, JDP2, and JUN. However, the consequences of ATF4 interaction with these bZIP factors, except for JPD2, have not been extensively investigated.

For instance, siRNA knockdown of JDP2 leads to the upregulation of several ATF4 target genes, including DR4, DR5, and ASNS, and increases the sensitivity of cells to tumor necrosis factor-related apoptosis-inducing ligand (TRAIL), thereby promoting apoptosis. JDP2 is also known to be involved in cellular senescence. Nakade and colleagues in a model of replicative senescence, showed that JDP2 can upregulate the expression of cell cycle inhibitor genes p16INK4a and p19ARF. This upregulation is proposed to be mediated by the inhibition of the recruitment of polycomb repressive complexes to the promoters of these genes, as suggested by the authors [178].

The transcription factor FOS is essential for normal cell proliferation and differentiation, and its elevated levels are often observed in various types of cancer, classifying FOS as a proto-oncogene [179]. Seshadri and Campisi demonstrated that in senescent late-passage human fibroblasts, the expression of FOS is significantly reduced, being approximately tenfold lower compared to young fibroblasts. Additionally, FOS in senescent cells exhibits minimal responsiveness to serum stimulation, in contrast to its inducibility in young cells [180]. Conversely, senescent T-cells exhibit high levels of FOS, which is distinct from naïve T-cells. In naïve T-cells, FOXO1 maintains them in a quiescent state by downregulating the expression of FOS and FOSB transcription factors [181].

The transcription factor MAF plays a significant role in regulating pluripotency, proliferation, and self-renewal genes in human ASCs. Chen and colleagues demonstrated that hASCs lack MAF during replicative and ROS-mediated senescence, which is associated with reduced osteogenic differentiation [182].

Fos-related antigen 1 (FRA1) is a key bZIP TF that contributes significantly to the activity of AP-1 and plays crucial roles in processes like cell differentiation, proliferation, and cancer progression. In a study by Yang and colleagues, it was observed that FRA1 expression significantly increases during angiotensin II–induced senescence of rat aortic endothelial cells (RAEC) and in the arteries of Ang II–infused mice. When FRA1 was knocked down using siRNA, it attenuated Ang II–induced senescence of vascular smooth muscle cells and RAEC in vitro. Furthermore, shRNA knockdown of FRA1 alleviated the Ang II–induced vascular aging phenotype in vivo. The study also revealed that in senescent cells, FRA1 forms a complex with JUN and binds to the promoter regions of cell cycle inhibitors p16 and p21, leading to their upregulation [183]. This research highlights the significant role of FRA1 in cellular senescence and vascular aging [183].

Indeed, all the mentioned bZIP transcription factors play pivotal roles in cell physiology and senescence. However, the precise cellular impact of dimeric complexes formed by ATF4 with most of these bZIP factors remains elusive. Therefore, further investigation of the effects of these dimers is of utmost importance for gaining a comprehensive understanding of cellular senescence pathophysiology. Moreover, such research may potentially unveil new opportunities for the development of novel senotherapeutic compounds aimed at targeting these intricate regulatory mechanisms.

A summary of literature reporting the impact of some known ISR transcription factors and their partners in senescence is presented in Table 1.

### 7.5. ATF4 and TRIB3

The pseudokinase Tribble 3 (TRIB3 also known as TRB3, NIPK, and SKIP3) is a member of the Tribble family of serine-threonine pseudokinases, characterized by structural differences from canonical protein kinases, such as the absence of the typical Mg^2+^-binding DFG domain and a glycine-rich ATP-binding loop. TRIB3 is involved in various signaling pathways, including those related to cell differentiation, proliferation, and organism development [184,185]. Researchers are increasingly focusing on the significant activation of TRIB3 under various stress conditions, including hypoxia and endoplasmic reticulum stress. Overexpression of the TRIB3 gene is also observed in many tumor cells [184,186,187].

The possibility of human TRIB3 and ATF4 binding was first discovered by Bowers and colleagues in a yeast two-hybrid assay [187]. Although the direct formation of the TRIB3-ATF4 complex has not been confirmed, Örd and colleagues found through ChIP-Seq analysis that TRIB3 predominantly resides at the same chromatin sites as ATF4 and constraints ATF4 activity [188].

Ohoka and colleagues discovered that ATF4, especially in a heterodimeric complex with CHOP, can stimulate the expression of the TRIB3 gene. They also found that TRIB3 acts as an effector molecule in a negative feedback loop for ATF4, repressing the transcriptional activity of both ATF4 and CHOP [189]. Subsequently, other researchers confirmed this finding but did not emphasize which dimeric complex of ATF4 primarily activates TRIB3 expression [190,191]. Örd and colleagues propose that TRIB3 plays a cytoprotective role by rescuing cells from programmed cell death when ATF4 is overexpressed [192].

Indeed, it is intriguing that while CEBPB can enhance the expression of the TRIB3 gene, the TRIB3 kinase in turn inhibits the transcriptional activity of CEBPB [193,194]. However, the precise role of TRIB3 in cellular aging remains poorly understood. On the one hand, Li and colleagues have reported that TRIB3 promotes the progression of acute promyelocytic leukemia by preventing the degradation of the oncogenic protein PML-RARα and that TRIB3 depletion leads to p53-induced senescence [195]. On the other hand, Gu and colleagues have shown that silencing the TRIB3 gene in chondrocytes from osteoarthritis patients induces autophagy in these cells. This was indicated by increased levels of LC3-II and BECN1 and decreased levels of p62, the main autophagy regulator proteins. Furthermore, this autophagic response was accompanied by a decrease in the number of SA-βGAL-positive cells and reduced levels of the cell cycle inhibitors p16 and p21, suggesting a mitigation of senescence processes [196].

The regulatory activity of TRIB3 appears to be context-specific, depending on the stressful condition. The findings of Corcoran and colleagues suggest that tunicamycin or thapsigargin-induced ER stress increases TRIB3 protein levels, whereas p53 and oncogenic stress decrease TRIB3 levels [197]. Finally, Wang and colleagues shed light on the pathogenic consequences of the interaction between TRIB3 and ATF4. They showed that TRIB3 attenuates lung fibrosis by negatively regulating ATF4 in lung cells and inhibits lung fibroblast activation by regulating ATF4 expression. This leads to the maintenance of normal epithelial-to-mesenchymal transition, suppression of extracellular matrix protein synthesis, and prevention of fibroblast transformation into myofibroblasts [198].

In summary, the intricate network of ATF4 dimerization partners within the C/EBP transcription factor family and other associated molecules provides a diverse landscape for understanding the development of senescence and cellular responses to stress. In particular, the ATF4-C/EBP complex has been shown to play a role in regulating the CDKN2A gene expression and influencing cell proliferation. In addition, the interaction between ATF4 and CEBPG appears to have an anti-senescent effect, whereas its association with CHOP promotes a pro-senescent outcome. Beyond C/EBP interactions, the ATF4-NRF2 complex emerges as a key player in cellular redox balance and anti-senescent processes. NRF2, a master regulator of the cellular oxidative stress response, can form a complex with ATF4, potentially regulating senescence-related gene expression. The ATF4-TRIB3 interaction adds another layer of complexity to our understanding of senescence regulation. TRIB3, acting as both an effector molecule and a regulator of ATF4, has context-specific effects on senescence, depending on the nature of the stressor and particular cell environment.

An important feature of ISR signaling is its ability to lead to different outcomes for cells—either apoptosis or pro-survival. The general mechanism that governs the choice between these outcomes remains unknown. However, since ATF4 is a key effector of the ISR, it can be hypothesized that different combinations of ATF4 partners in heterodimerization may alter the repertoire of ATF4-targeted genes. According to a literature analysis, approximately 17 bZIP transcription factors can form complexes with ATF4, and this diversity of partners illustrates the complexity of ATF4 regulation [94]. A question remains as to the extent to which different ATF4 dimerization partners exert control over specific target genes. In addition, it remains uncertain whether alterations in the levels of these transcription factor partners, triggered by the activation of other signaling pathways, can influence this process. Further research in this area promises to provide additional insights into the regulation of senescence and may lead to the development of novel senotherapeutic approaches.

## 8. Modulators of ISR and Their Impact on Senescence-Related Conditions

To date, several chemical compounds that modulate the ISR have been developed, some of which have shown effects on cellular senescence. The impact of the well-known ISR modulators on the course of cellular senescence is depicted in Figure 3.

### 8.1. Inhibitors of ISR

One such compound, ISRIB, inhibits ISR by facilitating the assembly of two heterotetrameric subcomplexes, eIF2Bβγδε, and the eIF2Bα2 homodimer, forming a functional decameric complex known as eIF2B. However, at high concentrations of p-eIF2α, the effect of ISRIB is significantly reduced because almost all newly formed eIF2B complexes quickly bind to p-eIF2α [199]. In an experiment using a silicosis lung model in C57BL/6 mice, Li and colleagues demonstrated that ISRIB promotes the recovery of lung function. This is evidenced by reduced collagen accumulation, myofibroblast generation, and decreased senescence of alveolar cells, as indicated by lower levels of p16, p21, p-p53, and PAI-1 [200].

The anti-senescent effect of the PERK kinase inhibitor GSK2606414 and ISRIB has been demonstrated in hypothalamic neural stem cells. These cells are recruited from the hypothalamus to the site of injury caused by the growth of adamantinomatous craniopharyngioma. Under the influence of increased levels of oxidized low-density lipoproteins caused by the injury, these cells undergo senescence and the ISR pathway is also activated. Suppression of the ISR with GSK2606414 or ISRIB results in the attenuation of their senescent state [201]. However, it should be noted that GSK2606414 and GSK2656157, which were initially positioned as highly specific to PERK, have subsequently been found to inhibit RIPK1 and suppress RIPK1-dependent TNF-mediated cell death [202].

Compound C16, an oxindole/imidazole derivative, acts as a selective inhibitor of PKR kinase and can suppress proliferation and angiogenesis in HCC cells [203]. However, to date, the role of Compound C16 in cellular senescence remains poorly understood.

### 8.2. Activators of ISR

Salubrinal is an ISR activator as it inhibits two regulatory subunits of protein phosphatase 1, namely CReP and GADD34, which are responsible for dephosphorylating eIF2α [204]. Additionally, salubrinal can directly interact with the Bcl2 protein, maintaining it in a functionally active state and thereby blocking cell apoptosis [205]. Li and colleagues demonstrated that H_2_O_2_-induced senescence in mouse bone marrow mesenchymal stem cells (BMSCs) is accompanied by the activation of ISR signaling. They found that this senescent phenotype could be alleviated by salubrinal treatment. In cultures of senescent BMSCs, salubrinal reduced the number of SA-βGAL-positive cells by 34.8% and significantly enhanced the proliferative activity of approximately 28.5% of the cells. However, exposure to salubrinal also led to an increase in the number of early apoptotic cells by 6.2% [206]. Although salubrinal maintains the activity of the Bcl2 protein, which normally protects cells from apoptosis, this pro-apoptotic effect is likely to be a result of excessive ISR activation. Interestingly, in the same experiment, treatment of BMSCs with AMG44, a PERK kinase inhibitor, suppressed ISR signaling but did not affect the senescent state of these cells [206]. The anti-senescent effect of salubrinal has been demonstrated not only in a model of H_2_O_2_-induced senescence but also in a more complex system where senescence was induced by suppressing the expression of the autophagy receptor gene NBR1 using siRNA. In various cell lines, the abrogation of NBR1 led to the development of cellular senescence through the promotion of p38 activity, oxidative stress, and ER stress. In this model, the knockdown of PERK or IRE1α increased the number of senescent cells, while salubrinal attenuated this process. Thus, activation of the ISR signaling reduced senescence in this model [206].

Halofuginone, a halogenated synthetic derivative of the alkaloid febrifugine extracted from the root of the plant *Dichroa febrifuga*, has a broad spectrum of activities. Among its effects, it stimulates the ISR by inhibiting glutamyl-prolyl tRNA synthetase (EPRS) and subsequently activating the kinase GCN2 due to proline deficiency. Interestingly, the effect of halofuginone can be reversed by adding exogenous proline or by activating EPRS [207]. Pitera and colleagues discovered that high doses of halofuginone lead to the development of an atypical ISR in HeLa cells. This atypical ISR is characterized by a high degree of eIF2α phosphorylation but a suppression of ATF4 proteins. The authors suggest that this phenomenon may be related to emerging defects in elongation and ribosomal stalling during protein synthesis due to proline deficiency [208]. Tsuchida and colleagues performed a high-throughput chemical library screening of 5861 compounds, measuring luciferase reporter activity in A549 lung adenocarcinoma cells using a promoter with an NRF2-binding ARE sequence. It is worth noting that these cells carry a homozygous KEAP1 loss-of-function mutation, which results in the constant accumulation of NRF2 in the nucleus. They identified febrifugine and SH-168 as the most potent inhibitors, suppressing NRF2 activity by more than 70%. They also included halofuginone, a less toxic derivative of febrifugine, in the study and found that halofuginone effectively inhibited NRF2 accumulation by activating the ISR pathway and subsequently blocking global protein synthesis. However, halofuginone sensitized cancer cells to a greater extent than immortalized normal epithelial cells, highlighting its differential effect [209]. Nevertheless, this suggests that the effect of halofuginone to some extent mediates the suppression of NRF2 translation, which is one of the key anti-senescent molecules.

A distinct category of ISR modulators includes activators of four different kinases responsible for phosphorylating eIF2α: PKR, GCN2, PERK, and HRI.

BTdCPU, a compound belonging to N,N′-diarylureas, was identified as a novel activator of the HRI kinase by Chen and colleagues. Its major advantage is that it does not induce oxidative stress and has strong specificity for HRI [210].

Ganoderic acid D (GA-D), a highly oxygenated natural tetracyclic triterpenoid found in Ganoderma lucidum, was discovered by Xu and colleagues to prevent senescence in human amniotic mesenchymal stem cells (hAMSCs). This was achieved by upregulating the expression of PERK and NRF2 and promoting the intranuclear transfer of NRF2 in senescent cells. Interestingly, the addition of the PERK inhibitor GSK2656157 to hAMSC cultures mediated the suppression of PERK-NRF2 signaling, exacerbating the senescent state of these cells. These findings suggest that GA-D alleviates hAMSC senescence through activation of the PERK/NRF2 signaling pathway and may be a promising candidate for anti-aging drug discovery [211].

In summary, there is currently a wide range of ISR modulators available and their potential in the field of senotherapy is highly promising. These compounds have diverse activities, ranging from activation to inhibition of the ISR signaling, and play a crucial role in regulating fundamental cellular processes, including senescence. The complex interactions between different components of the ISR, such as kinases, transcription factors, and downstream effectors, provide a diverse landscape for potential therapeutic interventions.

For instance, compounds such as the ISR inhibitor ISRIB have shown the ability to alleviate cellular stress responses and senescence in specific cell types. On the other hand, salubrinal, an ISR activator, affects cellular fate decisions related to senescence and apoptosis through modulation of Bcl2 activity. In addition, researchers have identified several compounds that selectively target ISR-related kinases such as PERK, HRI, and PKR. These compounds have shown the potential to influence cellular senescence, paving the way for novel senotherapeutic strategies. In addition, compounds such as halofuginone and GA-D have emerged as notable ISR modulators. They have demonstrated the capability to regulate NRF2 signaling and thereby affect oxidative stress and senescence processes. These findings hold great promise for the development of innovative anti-aging agents.

In conclusion, the current understanding of ISR signaling and its intricate network of modulators offers exciting prospects for senotherapy. These compounds, with their ability to finely regulate cellular responses to stress and senescence, may ultimately lead to the creation of groundbreaking senotherapeutic drugs in the near future.

## 9. Take Home Message

This review has offered a distinctive perspective on the integrated stress response pathway and its intricate connection with cellular senescence. While prior research has explored the role of ISR in aging and stress responses, our review provides novel insights by focusing on the pivotal role of ATF dimers within this pathway. The core concept of our work aimed to highlight the importance of investigating diverse ATF4 heterocomplex variants. Our rationale is rooted in the observation that, presently, a substantial portion of experimental studies overlook this aspect, potentially contributing to the disparities observed in the outcomes.

## 10. Conclusions

The integrated stress response is a fundamental adaptive mechanism utilized by cells in response to a variety of cellular stressors, however, its role is not limited to stress response -this pathway is a double-edged sword in cellular senescence, contributing to both its alleviation and persistence. In this review, we have covered most of the components of the ISR pathway and emphasized the significance of ATF dimers, within the ISR pathway and their intricate role in cellular senescence. By exploring the dynamic interactions of ATF dimers with other transcription factors and the nuances of ATF dimerization, we provided unique insights into the regulatory mechanisms governing senescence. We hypothesize that senescence development involves a specific combination of ISR transcription factor complexes, influencing the senescence phenotype. Understanding these complexes could lead to precise senescence therapies and novel ISR modulators, offering insights into age-related disease treatment.

## Figures and Tables

**Figure 1 ijms-24-17423-f001:**
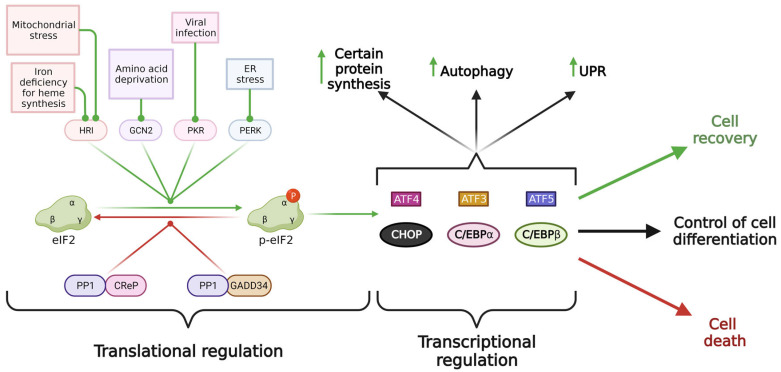
Molecular mechanism of the integrated stress response. The exposure to various stressors leads to the activation of one or more kinases (HRI, GCN2, PKR, PERK) which phosphorylate translation initiation factor eIF2α, resulting in the inhibition of global protein translation but stimulating the synthesis of certain molecules such as the transcription factors ATF3, ATF4, ATF5, CEBPA, CEBPB and CHOP. These factors can form heterodimers through the bZIP domain then bind to DNA targets and activate the expression of genes involved in cellular adaptation. Interestingly, ATF4 can also be expressed in the absence of stress through a mTORC1-dependent mechanism, thereby stimulating the synthesis of specific proteins. Once the stress has been resolved, the ISR pathway is terminated by p-eIF2α dephosphorylation mediated by Protein Phosphatase 1 (PP1) complexing with CReP or GADD34. By inhibiting global protein synthesis, ISR can maintain cells in a state of quiescence, thereby regulating their differentiation and proliferation. In conditions of exposure to excessive stress, ISR can induce programmed cell death. Thus, ISR is a highly sensitive mechanism that finely regulates cellular metabolism and determines cell fate. Green arrows indicate improvement and red arrows indicate deterioration.

**Figure 2 ijms-24-17423-f002:**
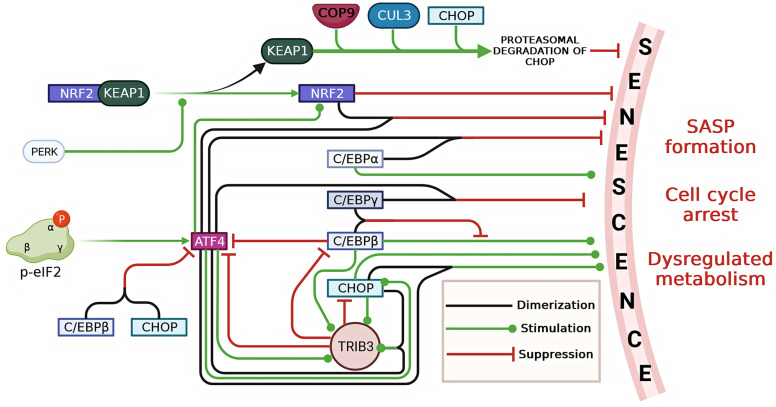
ATF4-binding partners network in the context of cellular senescence. The major effector transcription factor of the ISR signaling is ATF4. Since ATF4 is a bZIP-containing transcription factor, it can exert its transcriptional activity exclusively as a part of a heterodimeric complex with a few other transcription factors. Moreover, the ATF4 binding partner is capable of determining the transcriptional activity of the entire dimeric complex. Given that ATF4 homodimers are unstable, it is important to determine the effect of a particular binding partner on ATF4. In this figure, we have demonstrated the multifaceted effects of ATF4 on cellular senescence using its binding partners NRF2, CHOP, CEBPA, CEBPB, CEBPG, and TRIB3 as examples. Green lines indicate senescences stimulation, red arrows indicate senescence suppression and black lines connect heterodimeric partners.

**Figure 3 ijms-24-17423-f003:**
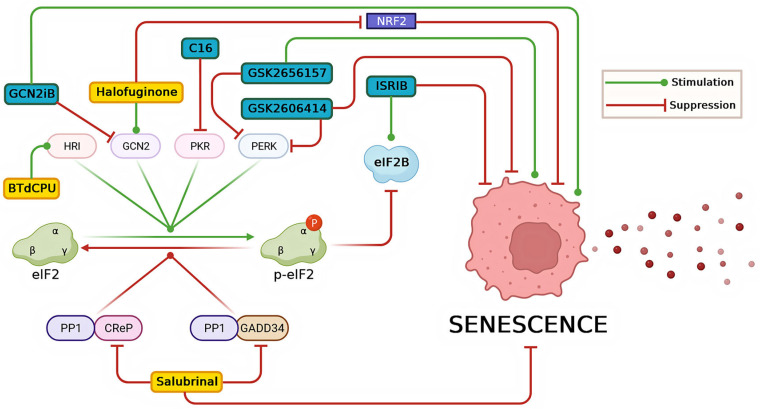
Effects of different ISR modulators on cellular senescence. To date, several compounds have been developed to modulate ISR signaling. The yellow boxes display compounds that stimulate ISR, and the blue boxes show agents that inhibit ISR. The diversity of these compounds allows for the modulation of ISR at different stages. For example, the compounds can act both to inhibit (GCN2iB, C16, GSK2606414, GSK2656157) or activate (BTdCPU, halofuginone) ISR kinases, suppress dephosphorylation of p-eIF2α (salubrinal) or stimulate the nucleotide exchange factor eIF2B (ISRIB). Some of these agents have been reported to modulate cellular senescence as well as senescence-related conditions. ISR modulators therefore offer great potential for the development of novel senotherapeutics. Green lines indicate senescences stimulation and red arrows indicate senescence suppression.

**Table 1 ijms-24-17423-t001:** Impact of ATF4 dimerization partners on senescence.

Transcription Factor	Interacting Partner	Effect on Senescence	References
ATF4	CEBPA, CEBPG	Senescence alleviation through regulation of CARE-containing genes	Horiguchi, M. et al. (2012) [158]; Huggins, C.J. et al. (2016) [159]; Huggins, C.J. et al. (2013) [160]
	CHOP	Stimulation of pro-senescence protein p21 expression	Inoue, Y. et al. (2017) [165]
	NRF2	Senescence alleviation presumably through the expression of NRF2-target genes, which exert an anti-senescence effect	He, C.H. et al. (2001) [170]
ATF3	Not stated	Stimulation of replicative senescence through upregulation of pro-senescence proteins like p16 and p21	Zhang, C. et al. (2021) [139]
		ATF3 knockdown results in aggravated senescence in macrophages exposed to Pseudomonas aeruginosa	Zhao, Q. et al. (2021) [140]
		Aggravation of acrylamide-induced senescence by upregulation of p53 and p21 pro-senescent proteins in macrophages	Kim, K.-H. et al. (2015) [141]
ATF4	Not stated	Upregulation of pro-senescence protein p16 in senescent renal tubular epithelial cells	Liu, J. et al. (2015) [146]
		Upregulation of pro-senescence protein p16 in oxidative stress induces senescence in MEFs	Sakai, T. et al. (2019) [147]
CHOP	Not stated	Senescence aggravation in alveolar epithelial cells from patients with idiopathic pulmonary fibrosis through enhancement of ROS generation and activation of the NF-κB pathway—factors promoting senescence	Jing, X. et al. (2022) [166]
CEBPA	Not stated	Senescence aggravation by upregulation pro-senescence factors: CXCL1, CXCL5 and CXCL15	Guan, Y. et al. (2020) [162]
JDP2	Not stated	Aggravation of MEFs replicative senescence by upregulation of p16 and p19 proteins	Nakade, K. et al. (2009) [178]
FOS	Not stated	FOS expression is significantly increased during fibroblast replicative senescence	Seshadri, T. et al. (1990) [180]
	Not stated	FOS is upregulated in senescent T-cells, whether FOXO1 ensures the naïve state of T-cells through downregulation of FOS and FOSB	Delpoux, A. et al. (2021) [181]
MAF	Not stated	MAF disappears in senescent hADMSCs, leading to reduced osteogenic differentiation capacity	Chen P.-M., et al. (2015) [182]

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
