# Peer review of "Integrated Stress Response (ISR) Pathway: Unraveling Its Role in Cellular Senescence"

_ijms, 2023, doi:10.3390/ijms242417423_

Round 1

Reviewer 1 Report

Comments and Suggestions for Authors

This review paper's authors have compiled a fairly comprehensive overview of the current knowledge of Integrated Stress Response (ISR) in Cellular Senescence. The text summarizes the results and findings of 200+ research papers and includes three summary figures that provide information for molecular modulators and mechanisms involved in cellular senescence. 

Overall, there are only a few relatively minor areas for improvement that could further improve this review paper:

1) Some of the sentences or text paragraphs seem to be too long. For example, between lines 30-40, one very long sentence could be rearranged into 3 to 4 (or more) shorter sentences and make the text more accessible to digest by readers of the journal. 

2) Some of the text for the used abbreviations is provided directly in the text, but for many of the used abbreviations, no information is included anywhere in the text. For example, CHOP is mentioned 20+ times in the manuscript, but in none of those instances is provided that CHOP stands for "C/EBP homologous protein". One easy solution for these issues is adding a table in the paper that will describe all abbreviations used in the main text. Such a table or list of used abbreviations could also help to shorten the text in some of the manuscript sections.

3) Authors stated multiple times that there is an absence of specific markers for senescent cells. However, there are molecular markers that are routinely used in research studies to quantify or detect senescence (for example, please see this "10 Must-have Markers for Senescence Research": https://blog.cellsignal.com/10-must-have-markers-for-senescence-research). Indeed, using these known markers is challenging for the universal detection of senses due to their nonspecificity and the existence of different senescence programs. The authors could provide some clarification (or an entire paragraph/section) to describe the most commonly used markers and their limitations to justify (back up) the statement that there is an "absence of specific markers for senescent cells (as mentioned in lines 9, 46, 127, etc.)".

4) One minor missing part of the study is to provide details on tissue specificity of senescent cell accumulation (which organs/tissues are most and least prone in the magnitude of senescence?). For example, there are reports of a positive association between senescence and age within the kidney tissues and others. While other tissue types, such as adipose, gut, prostate, and thymus, have been reported to have senescence that is not significantly associated/correlated with age (expand on reference #11).

Comments on the Quality of English Language

Overall, excellent quality of English language writing. Some sentences appear too long and too descriptive in the minute details of cited studies. That could be both helpful and overwhelming for readers. Generally, the review papers should not be too granular in the details of the cited studies. They should provide predominantly a high-level summary of previously reported findings, mechanisms, and paradigms.

Author Response

Response to Reviewer 1

We are very grateful to Reviewer for valuable and logic remarks. We carefully followed his instructions to improve our manuscript. The following is our answers to the remarks of reviewer.

Remark 1

Some of the sentences or text paragraphs seem to be too long. For example, between lines 30-40, one very long sentence could be rearranged into 3 to 4 (or more) shorter sentences and make the text more accessible to digest by readers of the journal. 

Response:

We appreciate the reviewer’s feedback on the readability of our manuscript. We have revised the text between lines 30-40, breaking down the longer sentences into shorter, more digestible segments. Additionally, we have implemented this restructuring throughout the manuscript to improve clarity and ease of reading. All changes have been highlighted in yellow to facilitate easy identification of the modifications made in response to the reviewer's comments (Briefly lines 67-76; 323-326; 358-362; 395-400; 511-515; 529-533; 547-550; 575-581; 588-592).

Remark 2

Some of the text for the used abbreviations is provided directly in the text, but for many of the used abbreviations, no information is included anywhere in the text. For example, CHOP is mentioned 20+ times in the manuscript, but in none of those instances is provided that CHOP stands for "C/EBP homologous protein". One easy solution for these issues is adding a table in the paper that will describe all abbreviations used in the main text. Such a table or list of used abbreviations could also help to shorten the text in some of the manuscript sections.

Response:

Thank you for pointing out the oversight regarding the use of abbreviations in our manuscript. We agree that a comprehensive list of abbreviations would aid reader comprehension. In response, we have compiled a list of all abbreviations and have incorporated it into the manuscript (Lines 26-62). We trust this addition will make the manuscript more reader-friendly and accessible.

Remark 3

Authors stated multiple times that there is an absence of specific markers for senescent cells. However, there are molecular markers that are routinely used in research studies to quantify or detect senescence (for example, please see this "10 Must-have Markers for Senescence Research": https://blog.cellsignal.com/10-must-have-markers-for-senescence-research). Indeed, using these known markers is challenging for the universal detection of senses due to their nonspecificity and the existence of different senescence programs. The authors could provide some clarification (or an entire paragraph/section) to describe the most commonly used markers and their limitations to justify (back up) the statement that there is an "absence of specific markers for senescent cells (as mentioned in lines 9, 46, 127, etc.)".

Response:

Thank you for your insightful comment regarding the statement on the absence of specific markers for senescent cells in our manuscript. We acknowledge the existence of molecular markers commonly used in senescence research. Our intent was not to overlook these markers but to emphasize the challenges associated with their specificity and universality in detecting senescent cells.

In light of your suggestion, we propose to add a new paragraph (lines 125-162) in our manuscript that delves into the most commonly used markers in senescence research, such as p16^INK4a, p21^CIP1/WAF1, and SA-β-gal. This addition provide a overview of these markers, discussing their roles and the extent to which they are utilized in the field.

By including this paragraph, we aim to justify our statement about the absence of specific markers for senescent cells, underlining the complexity and the current limitations in the field of senescence research. We believe this addition will enhance the manuscript's accuracy and provide readers with a clearer understanding of the challenges in identifying senescent cells.

Remark 4

One minor missing part of the study is to provide details on tissue specificity of senescent cell accumulation (which organs/tissues are most and least prone in the magnitude of senescence?). For example, there are reports of a positive association between senescence and age within the kidney tissues and others. While other tissue types, such as adipose, gut, prostate, and thymus, have been reported to have senescence that is not significantly associated/correlated with age (expand on reference #11).

 Response:

Thank you for highlighting the necessity to address tissue specificity in the context of senescent cell accumulation. In response to your comment, we have carefully revised the relevant sentence pertaining to reference #11 (line 80) to more accurately reflect the tissue-specific accumulation of senescent cells. We believe these amendments substantially improve the manuscript by providing a clearer, more comprehensive picture of the differential accumulation of senescence across tissues.

Reviewer 2 Report

Comments and Suggestions for Authors

I am grateful for the opportunity to review this manuscript. The authors have conducted a literature review on the role of the Integrated Stress Response (ISR) in cellular senescence.

The authors have done a good job of synthesising and integrating the information. This field of study is not novel, but the in-depth analysis of the mechanisms of cellular ageing is an area of great interest.

The main transcription factors in these processes and their effects are reviewed. 

However, I consider that the present paper does not present novel or different data from other reviews in the same line and, therefore, I do not believe that it is of particular interest to readers. I recommend that they use a different type of methodology in which to collect the data, so that novel results can be presented.

Best regards,

Author Response

Response to Reviewer 2

We are very grateful to the Reviewer for taking the time to review our manuscript. We appreciate your feedback and the opportunity to address your concerns.

We understand your comments regarding the novelty of our work. However, we believe that our review offers several significant contributions to the field, including:

Unique Perspective: Our review provides a distinct perspective on the relationship between the Integrated Stress Response (ISR) pathway and cellular senescence, offering a comprehensive analysis of this complex interplay. We discuss possible ISR modulators and their impact on senescence-related conditions, providing a foundation for the development of targeted anti-senescence therapies based on ISR modulation.

Role of ATF4 and Partners: We have thoroughly examined the central role of ATF4, the primary transcription factor activated by ISR, and its interactions with other factors in mediating the stress response and senescence. This discussion sheds light on the intricate molecular mechanisms involved.

 Importance of Dimerization: We emphasize the importance of considering ATF4's dimerization with other transcription factors for a comprehensive understanding of its role in senescence, highlighting the intricate nature of this pathway. We discuss how outcomes of the ISR pathway can vary depending on the presence of specific ATF4-interacting proteins, demonstrating the context-dependent nature of these interactions

Implications for Cellular Metabolism: Our work explores the implications of ISR and senescence on cellular metabolism, offering valuable insights into this critical aspect of cellular physiology.

Up-to-Date References: We would like to highlight that our review incorporates the most recent research findings, as evidenced by the inclusion of 10 references from 2023, 18 from 2022, 20 from 2021, and 15 from 2020, out of a total of 206 references. This accounts for approximately 30 percent of our reference list, underscoring the timeliness and relevance of our work.

We believe that these aspects contribute to the significance of our review in the context of existing literature. Moreover, to date, not many papers have been published on the topic of the integrative stress response and senescence. The closest-by-topic and date paper is the review article titled "Modulating the Integrated Stress Response to Slow Aging and Ameliorate Age-Related Pathology" by Derisbourg, Hartman, and Denzel, published in Nature Aging in September 2021. However, this work focuses primarily on the relationship between the Integrated Stress Response (ISR) and aging-related processes. While this review shares a common theme of the ISR pathway with your work, there are some key differences: The Derisbourg et al. review is centered on aging and age-related pathologies. It explores how modulating the ISR pathway can potentially slow down the aging process and mitigate age-related diseases. This is the primary focus of their review. And this review does not delve into the specific details of ATF dimers within the ISR pathway. It may not discuss the nuances of ATF interactions and their implications for cellular senescence in depth.

We understand your recommendation to explore different methodologies for data collection, and we will certainly consider this for future research. However, we also believe that synthesizing and analyzing existing knowledge is a valuable endeavor, as it provides a comprehensive overview of the current state of the field.

Once again, we appreciate your thoughtful review, and we hope that the unique insights presented in our review will be of interest to the readers of the International Journal of Molecular Science.

Round 2

Reviewer 2 Report

Comments and Suggestions for Authors

I am grateful for the explanation provided by the authors. However, no changes have been made to improve the quality of the manuscript.

I recommend that you review the methodology in order to present a paper that is of interest to the reader.

Author Response

Response to Reviewer 2

Reviewer’s remark:

I recommend that you review the methodology in order to present a paper that is of interest to the reader

Answer:

We are very grateful for reviewer’s attention to our manuscript. Considering the fact that the review methodology could not be significantly revised at this stage, we tried to enhance our manuscript by adding some visual supplements.

We included Table 2 in the manuscript. This table refers to a number of experimental works demonstrating the contribution of ATP4 partners to the development of cellular senescence.

In addition, we have expanded the list of abbreviations, highlighting separately the general subjects as specific protein factors mentioned in the text (Table 1. Protein abbreviations/gene names).

We have also shortened the conclusion in order to achieve a clearer understanding of what we wanted to say with our review.

We hope that these improvements will contribute to better reader understanding.
